# The rising burden of female cancer in Ethiopia (2000–2021) and projections to 2040: Insights from the global burden of disease study

Molalign Aligaz Adisu[1]*, Tesfaye Engdaw Habtie[2], Tegene Atamenta Kitaw[2], Abraham Dessie Gessesse[1], Bogale Molla Woreta[2], Yabibal Asfaw Derso[2], Alemu Birara Zemariam[1]

**1** Department of Pediatrics and child health Nursing, College of Health science, Woldia University, Woldia, Ethiopia, **2** Department of Nursing, College of Health science, Woldia University, Woldia, Ethiopia

* molalignaligaz@gmail.com, molalign.a@wldu.edu.et

## Abstract

### Background

Female cancers—breast, cervical, ovarian, and uterine—pose significant public health and socio-economic challenges, particularly in low- and middle-income countries like Ethiopia. However, detailed and geographically disaggregated data are limited, hindering effective policymaking. To address this gap, our study utilizes the Global Burden of Disease (GBD) methodology to analyze 21 years (2000–2021) of national and sub-national trends and risk factors for these cancers in Ethiopia, with projections to 2040, to support targeted cancer control and health system strengthening.

### Methods

Using the 2021GBD data, we analyzed the national and sub-national prevalence, incidence, mortality, disability-adjusted life years (DALYs), years of life lost (YLLs), and years lived with disability (YLDs) for female specific cancer in Ethiopia. An Autoregressive Integrated Moving Average (ARIMA) model was employed for projecting epidemiological trajectories through 2040. All statistical analyses and data visualization were performed using Python.

### Results

In 2021, the Ethiopian incidence of female breast, cervical, ovarian, and uterine cancer was 7,308 (95% uncertainty interval (UI): 5,794–9,199), 7,884 (95% UI: 5,759–11,765), 2,054 (95% UI: 1,034–2,929), and 669 (95% UI: 422–1,126), respectively. Cervical cancer accounts for the highest number of DALYs, 162,776 (95% UI: 119,900–239,116), followed by breast, ovarian, and uterine cancer at 155,931 (95%

**Data availability statement:** The data used in this study accessed from to the GBD 2021 official website is available at https://ghdx.healthdata.org/gbd-2021.

**Funding:** The author(s) received no specific funding for this work.

**Competing interests:** The authors have declared that no competing interests exist.

UI: 123,015–196,249), 40,430 (95% UI: 19,885–57,414), and 8,882 (95% UI: 5,579–15,240), respectively. Projections to 2040 indicate a continued rise in incidence for all female cancers.

## Conclusions

Breast and ovarian cancers are emerging public health crises in Ethiopia, with significant increases in prevalence, incidence, and DALYs. While the cervical cancer burden is declining, rising YLDs indicate a growing need for long-term care. The projected rise in female cancer incidence calls for urgent, targeted interventions focused on early diagnosis, age-appropriate screening, and improved cancer care services to reduce the adverse impact on Ethiopian women's health.

## Introduction

Cancer remains a significant global health problem, the second leading cause of death worldwide, disproportionately impacting low- and middle-income countries where over 70% of cancer deaths occur [1,2]. Increasingly, the burden is most attributable to demographic change, epidemiologic transition, and continued inequities in access and quality of care [2,3]. In Sub-Saharan African countries, the convergence of a growing and ageing population, rapid urbanization, and lifestyle changes is driving a doubling of new cancer cases, further straining already compromised health systems [1,4].

Ethiopia, the second-most populous country in Africa, is currently facing a critical health transition. While tremendous progress has been made against infectious diseases, the country now grapples with a mounting double burden of disease, marked by a rapid increase in non-communicable diseases (NCDs), like cancer [5]. Female cancers, such as breast, cervical, ovarian, and uterine cancer represent a major public health challenge in this context [6,7]. These cancers collectively impose significant socio-economic burdens, resulting in premature mortality, impaired quality of life, and catastrophic healthcare expenditures on individuals and families, driving existing inequalities [8,9].

Despite the evident and growing threat, comprehensive, long-term, and granular data on the specific trajectory and geographic distribution of these critical female cancers in Ethiopia remain notably scarce. Existing national cancer registries are nascent or incomplete, and population-based incidence and mortality statistics are often limited, fragmented, or derived from disparate methodologies. This is a key data gap that considerably constrains policymakers, public health experts, and physicians' ability to accurately estimate the real size of the issue, identify high-burden areas, project future needs, and implement evidence-based national and sub-national cancer control policies. A robust understanding of disease trends over time and by administrative level is essential to effective resource planning, targeted intervention (e.g., screening, immunization, early treatment, and diagnosis programs), and the creation of responsive health policy [5,10].

This study employs the robust and validated framework the Global Burden of Disease (GBD) study to presents a comprehensive 21-year review (2000–2021) of national and sub-national prevalence, incidence, mortality, and disability-adjusted life years (DALYs), years of life lost (YLLs), years lived with disability (YLDs), and risk factor for breast, cervical, ovarian, and uterine cancer in Ethiopia. Additionally, it provides important projections of these burdens through 2040. This study is the first to provide GBD-based, multi-cancer, age- and region-stratified burden analysis and projections of female-specific cancer in Ethiopia. By systematically quantifying the evolving epidemiology of these cancers and projecting future trends of these cancers, the study addresses a critical evidence gap. The findings will provide invaluable information to the Ethiopian Ministry of Health, regional health bureaus, and global partners, enabling targeted cancer control strategies, strengthened health infrastructure, and ultimately reducing the adverse impact of female cancers on the health and quality of life of Ethiopian women. This research represents a significant contribution to global health literature by offering a model to understand and mitigate the rising cancer burden in resource-limited settings.

## Methods

### Data source and scope

This study employed estimates of the GBD Study from 2000 to 2021 to compare the incidence, prevalence, mortality, DALYs, YLDs, and YLLs for breast, cervical, ovarian, and uterine cancer in Ethiopia. We retrieved all data for this analysis from the GBD 2021 study, available through the Institute for Health Metrics and Evaluation (IHME) GBD Results Tool. The data was downloaded on May, 2025 using the following parameters: **Measures:** Prevalence, Incidence, Mortality, DALYs, YLLs, and YLDs; **Causes:** Breast cancer, Cervical cancer, Ovarian cancer, and Uterine cancer; **Locations:** Ethiopia (national and sub-national); **Sex:** Female; **Age:** all ages; and **Years:** 2000–2021. The findings are expressed in terms of rates per 100,000 populations with 95% uncertainty intervals (UIs). Data are accessed via Global Health Data Exchange (GHDx) at IHME, University of Washington (http://ghdx.healthdata.org/gbd-results-tool).

The GBD represents a systematic effort to quantify health loss due to diseases and injuries across various demographics and geographic locations. The data was gathered from a variety of sources, including national cancer registries, death records, and health care utilization data. Particular mortality rates and disease burdens were estimated with the Cause of Death Ensemble model and spatiotemporal Gaussian process regression.

A comprehensive analysis was conducted to assess national and sub-national patterns of cancer incidence and burden between 2000 and 2021. Specific search criteria included cancer type (breast, cervical, ovarian, and uterine), measure (incidence and mortality), metric (count and rate), place (Ethiopia), and sub-nations of the nine regions and two city administrations, and age group (all ages, age-standardized, and specific age groups). This approach facilitated the detection of high-burden sites and informed targeted public health interventions.

### Statistical analysis

Burden of female cancer was quantified as prevalence, incidence, mortality, YLLs, YLDs, and DALYs. Age-standardized rates (ASRs) for specific age groups, along with estimated values and 95% uncertainty intervals (UIs), were sourced from GBD 2021. ASR was determined according to the following formula:

$$ASR = \frac{\sum_{I=1}^{N} a_i w_i}{\sum_{I=1}^{N} w_i} * 100,000$$

Where $a_i$ represents the age-specific rate for the *ith* age group and $w_i$ denotes the number of individuals (or the weight) in the same age group within the GBD 2021 standard population. *N* is the number of age groups.

To analyze trends in cancer incidence and mortality, percentage changes were calculated by comparing counts from 2000 to 2021 using the formula:

$$Percentage\ of\ change = \frac{(Y_{2021} - Y_{2000})}{Y_{2000}} 100\%$$

Where Y refers to total incident cases or total deaths

Statistical analyses for trend and Autoregressive Integrated Moving Average (ARIMA) projection to 2040 were conducted using Python (version 3.12), using libraries *pandas* for data manipulation and *statsmodels* for ARIMA modeling [11]. Data visualization, including generation of plots illustrating trends and uncertainty intervals was done using the *matplotlib* and *seaborn* libraries [12].

### Ethics statement

The research did not need ethical board approval because it used a public dataset and conducted no human or animal experiments

## Results

### The national prevalence, incidence and death Trends of Female breast, cervical, ovarian and uterine cancer in Ethiopia (2000–2021)

In Ethiopia, the number of breast cancer increased significantly from 20,149 (95% UI: 15,063–27,411) cases in 2000–54,367 (95% UI: 44,141–66,730) cases in 2021, an increase of 169.82%. Similarly, the incidence of breast cancer increased from 2,935 (95% UI: 2,064–4,154) to 7,308 (95% UI: 5,794–9,199) new cases, an increase of 148.98%. For cervical cancer, prevalence rose from 24,610 (95% UI: 18,981–35,190) to 32,665 (95% UI: 23,296–49,268) cases (32.73% rise), while incidence rose more modestly from 7,573 (95% UI: 5,846–10,736) to 7,884 (95% UI: 5,759–11,765) new cases (4.10% rise). Ovarian cancer had a substantial increase in prevalence, from 2,793 (95% UI: 1,467–4,926) to 8,300 (95% UI: 4,214–11,948) cases (197.18% increase), and incidence, from 787 (95% UI: 394–1,391) to 2,054 (95% UI: 1,034–2,929) new cases (160.97% increase). Lastly, uterine cancer prevalence nearly doubled, from 2,037 (95% UI: 1,289–2,881) to 4,290 (95% UI: 2,660–7,363) cases (110.61% increase), with incidence increasing from 357 (95% UI: 230–511) to 669 (95% UI: 422–1,126) new cases (87.11% increase). Mortality trends revealed a significant increase in deaths from breast cancer, rising from 2,179 (95% UI: 1,543–3,068) in 2000–4,470 (95% UI: 3,589–5,584) in 2021, a 105.13% increase. Ovarian cancer deaths also saw a sharp rise, from 492 (95% UI: 229–857) to 1,211 (95% UI: 599–1,720) over the same period, a 146.12% increase. Uterine cancer deaths showed a notable increase from 213 (95% UI: 137–306) in 2000–316 (95% UI: 204–526) in 2021, a 48.35% increase. In contrast, cervical cancer was the only female cancer to demonstrate a declining mortality trend, with deaths decreasing from 5,080 (95% UI: 3,801–6,400) in 2000–4,485 (95% UI: 3,356–6,398) in 2021, a −11.71% decrease (Fig 1). These national estimates point to a widespread and increasing cancer burden on the Ethiopian health system.

### Sub-National prevalence, Incidence and mortality trends of breast, cervical, ovarian and uterine cancer in Ethiopia (2000–2021)

**Breast cancer.** The prevalence of breast cancer increased significantly in areas such as Oromia, rising from 8,903 (95% UI: 6,654–12,056) cases in 2000–22,810 (95% UI: 18,527–28,145) cases in 2021. Similarly, the number of new cases increased from 1,323 (95% UI: 928–1,865) to 3,243 (95% UI: 2,569–4,074). Amhara also demonstrated a significant increase, with incidence rising from 622 (95% UI: 435–886) to 1,570 (95% UI: 1,244–1,973) new cases and prevalence rising from 4,374 (95% UI: 3,275–5,947) to 11,855 (95% UI: 9,678–14,463) cases. Despite having a smaller

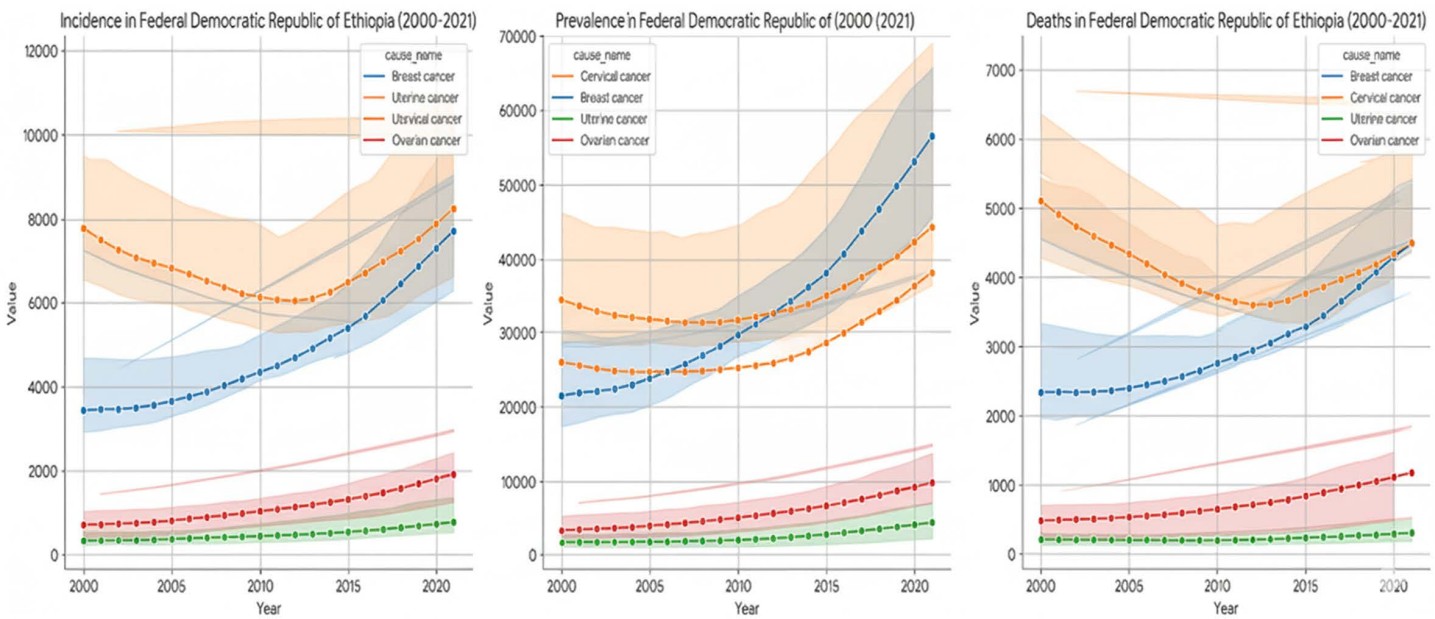

**Fig 1. Prevalence, incidence and mortality trends of breast, cervical, ovarian and uterine cancer in Ethiopia (2000-2021).**

geographic area, Addis Ababa showed a considerable burden, with incidence rising from 272 (95% UI: 191–386) to 665 (95% UI: 529–834) new cases and prevalence rising from 1,847 (95% UI: 1,380–2,504) to 4,964 (95% UI: 4,008–6,128) cases. Death trends for breast cancer mirrored the rise in prevalence and incidence. Oromia saw an increase in deaths from 742 (95% UI: 501–1075) in 2000–1,505 (95% UI: 1112–2013) in 2021, a 102.8% increase. Deaths in Amhara rose from 523 (95% UI: 317–843) to 965 (95% UI: 727–1270), an 84.5% increase. Addis Ababa also experienced a significant increase in deaths, from 185 in 2000–381 in 2021 (95% UI: 250–557), a 105.9% increase (Supplemental fig 1 in S1 File).

**Cervical cancer.** Cervical cancer trends also differed by region. In Oromia, incidence showed a slight increase from 3,514 (95% UI: 2,698–5,005) to 3,576 (95% UI: 2,586–5,321) new cases, while prevalence rose from 10,753 (95% UI: 8,290–15,361) to 12,027 (95% UI: 8,506–17,998) cases. Prevalence increased from 5,237 (95% UI: 4,028–7,515) to 6,108 (95% UI: 4,360–9,150) cases in Amhara, while incidence increased from 1,540 (95% UI: 1,180–2,185) to 1,575 (95% UI: 1,146–2,357) new cases. Prevalence increased from 4,394 (95% UI: 3,425–6,165) to 5,532 (95% UI: 3,923–8,429) cases in Southern Nations, Nationalities, and Peoples' Region (SNNPR) while incidence increased from 1,249 (95% UI: 981–1,732) to 1,304 (95% UI: 955–1,911) new cases. Death trend for cervical cancer showed a notable decline in Oromia and Amhara, but an increase in SNNPR. In Oromia, deaths decreased from 1,609 (95% UI: 1,123–2,410) in 2000–1,518 (95% UI: 1,033–2,312) in 2021, a −5.6% decrease. Amhara experienced a decrease from 1,306 (95% UI: 902–1,934) to 935(95% UI: 628–1,556) deaths, a −6.8% decrease. However, deaths in the SNNPR increased from 762 in 2000 (95% UI: 542–1,180) to 955 (95% UI: 645–1,441) in 2021, a 25.3% increase (Supplemental fig 1 in S1 File).

**Ovarian cancer.** Oromia once again had the highest burden of ovarian cancer, with incidence rising from 330 (95% UI: 164–581) to 828 (95% UI: 416–1,179) new cases and prevalence rising from 1,173 (95% UI: 615–2,060) to 3,363 (95% UI: 1,689–4,821). Amhara came next, with incidence increasing from 156 (95% UI: 78–275) to 433 (95% UI: 218–617) new cases and prevalence increasing from 552 (95% UI: 289–975) to 1,770 (95% UI: 897–2,537) cases. The percentage gains were significant even in smaller localities. The prevalence of ovarian cancer increased sharply throughout the country, for example, from 163 (95% UI: 86–287) to 488 (95% UI: 247–699) instances in Addis Ababa. On the other hand, the death rate observed in Oromia were increased from 165(95% UI: 48–299) in 2000–392 (95% UI: 140–601) in 2021,

a 137.6% increase. Deaths in Amhara rose from 138(95% UI: 53–245) to 312 (95% UI: 146–454), a 126.1% increase (Supplemental fig 1 in S1 File).

   **Uterine cancer.**  Regional increases were also seen in uterine cancer. In Oromia, incidence went from 153 (95% UI: 99–219) to 266 (95% UI: 168–447) new cases, while prevalence went from 866 (95% UI: 549–1,223) to 1,707 (95% UI: 1,059–2,925) instances. Incidence increased from 72 (95% UI: 47–104) to 132 (95% UI: 83–222) new cases, while prevalence increased from 409 (95% UI: 258–579) to 842 (95% UI: 522–1,440) cases in Amhara region. The deaths rate in Oromia rose from 73 (95% UI: 44–116) in 2000–108(95% UI: 66–188) in 2021, a 47.9% increase. In Amhara, deaths increased from 59(95% UI: 35–89) to 75(95% UI: 43–139), a 27.1% increase. The SNNP also saw an increase in deaths from 31 (95% UI: 19–48) to 61(95% UI: 37–105), a 96.8% increase (Supplemental fig 1 in S1 File).

**Deconstructing the total health loss: disability-adjusted life years, years lived with disability, and years of life lost for female breast, cervical, ovarian and uterine cancer in Ethiopia (2000–2021)**

   **Breast cancer: a steadily increasing total burden with significant disability.**  The DALYs attributable to breast cancer nationally increased from 80,211 (95% UI: 55,682–114,272) in 2000–155,931 (95% UI: 123,015–196,249) in 2021. This represents an absolute change of 75,719 and a significant 94.40% rise, highlighting the escalating comprehensive impact of breast cancer. Similarly, YLDs for breast cancer surged by an astounding 155.57%, from 1,723 (95% UI: 1,035–2,625) to 4,403 (95% UI: 2,897–6,261). This indicates that a significant and disproportionately growing share of the breast cancer burden is attributed to living with the disease and its sequel, suggesting increased long-term care and supportive needs. Although YLLs increased by 93.06%, from 78,488 (95% UI: 54,439–111,774) to 151,527 (95% UI: 119,191–190,520). The faster rise in YLDs compared to YLLs suggests that while mortality increased, there's also a growing population of breast cancer patients living longer with some form of disability (Fig 2).

   **Cervical cancer: decreasing mortality alongside modest rise in disability.**  Between 2000 and 2021, the national burden of cervical cancer in Ethiopia showed notable improvements in mortality outcomes accompanied by slight increases in disability. The total DALYs decreased by 15.22%, from 191,988 (95% UI: 147,614–271,797) to 162,776 (95% UI: 119,900–239,116). This decline was primarily driven by a substantial reduction in YLLs, which fell by 15.63%, from 189,435 (95% UI: 145,754–268,405) to 159,820 (95% UI: 117,577–234,935), reflecting lower mortality. Conversely, YLDs increased modestly by 15.78%, rising from 2,552 (95% UI: 1,736–3,661) to 2,955 (95% UI: 1,845–4,692), indicating a slight rise in the non-fatal health burden (Fig 2).

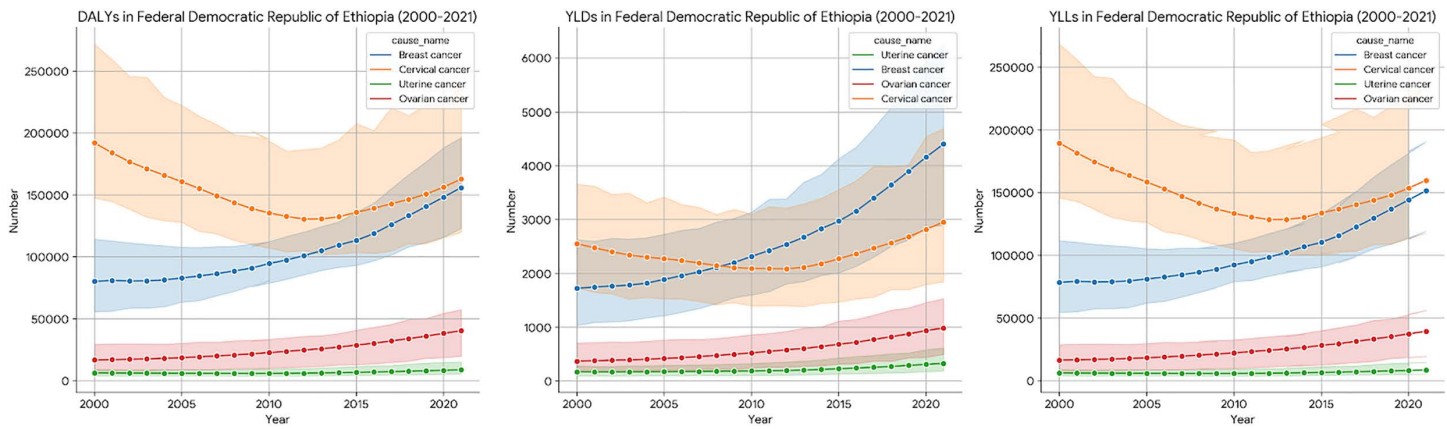

**Fig 2.  The DALYs, YLDs and YLLs trend of Female breast, cervical, ovarian and uterine cancer in Ethiopia (2000-2021).**

**Ovarian cancer: rapidly escalating disability burden.** The ovarian cancer DALYs soared by 140.68%, from 16,798 (95% UI: 8,359–29,466) in 2000–40,430 (95% UI: 19,885–57,414) in 2021, indicating a very significant and growing overall burden. The increase in YLDs was even more pronounced, with a 165.57% rise from 371 (95% UI: 180–709) to 985 (95% UI: 497–1,531). Additionally YLLs also increased substantially by 140.12%, from 16,427 (95% UI: 8,159–28,823) to 39,444 (95% UI: 19,445–56,039) (Fig 2).

**Uterine cancer: a growing disability contribution to the total burden.** The uterine cancer DALYs increased by 39.28%, from 6,377 (95% UI: 4,088–9,202) in 2000–8,882 (95% UI: 5,579–15,240) in 2021, signifying a growing total burden. Moreover, YLDs for uterine cancer rose sharply by 89.82%, from 173 (95% UI: 96–282) to 328 (95% UI: 189–616). Although, YLLs increased by 37.87%, from 6,204 (95% UI: 3,968–8,931) to 8,554 (95% UI: 5,385–14,586) (Fig 2).

### Sub-national burden dynamics: regional nuances and disparities

In Oromia, breast cancer has seen a significant rise in disease burden, with disability-adjusted life years (DALYs) increasing by 94.19%. Notably, years lived with disability (YLDs) surged by 155.38%, outpacing years of life lost (YLLs). Conversely, cervical cancer exhibited a decrease in DALYs by 7.83%, primarily due to an 8.34% reduction in YLLs, despite a 29.30% increase in YLDs. Amhara has similarly notable cancer burdens. Here, breast cancer DALYs rose by 68.44%, accompanied by a 122.45% increase in YLDs. In contrast, cervical cancer DALYs fell by 32.88%, with YLLs decreasing by 33.21% and YLDs showing a minor decline of 7.36%. Addis Ababa, though smaller in size, reported striking increases in YLDs for breast cancer (184.12%) and uterine cancer (114.05%). These figures may reflect enhanced diagnostic capabilities in urban settings, resulting in more identified cases that include disabilities. Furthermore, cervical cancer DALYs in Addis Ababa also dropped by 32.82%, driven by YLL declines. The SNNP region has experienced alarming increases across various cancer metrics, with ovarian cancer DALYs rising by 196.29%, YLLs by 195.67%, and YLDs escalating by 223.64% (Supplemental fig 2 in S1 File).

### Age-standardize prevalence, incidence and mortality rate of breast, cervical, uterine and ovarian cancer in Ethiopia

**Prevalence rates analysis.** Breast cancer prevalence rates were consistently the highest in age group 70+, with a notable rise from 561.72 per 100,000 populations in 2000 to 798.86 in 2021. Conversely, the 50–54 years age group had the highest cervical cancer incidence, which fell spectacularly from 384.40 per 100,000 populations in 2000 to 238.15 in 2021. For age group 60–64 years, ovarian cancer prevalence increased, from 55.80 per 100,000 populations in 2000 to 81.27 in 2021. Age group 65–69 years old had the highest uterine cancer prevalence, which increased from 78.12 per 100,000 in 2000 to 96.65 in 2021 (Fig 3).

**Incidence rates analysis.** The highest incidence of breast cancer was in the 70+ age group, which increased from 103.52 per 100,000 populations in 2000 to 144.32 in 2021. In the case of cervical cancer, the highest incidence changed; in 2000, the highest incidence was in the 60−64 age group, whereas in 2021, the highest incidence age group changed to 65−69, showing a huge decrease from 162.41 per 100,000 populations in 2000 to 91.72 in 2021. The highest incidence rates of ovarian cancer were observed in the 70+ age group, which increased from 22.74 per 100,000 populations in 2000 to 30.23 in 2021. The highest incidence of uterine cancer was also in the 70+ age group, but there was a slight decrease from 16.57 per 100,000 populations in 2000 to 16.53 in 2021 (Fig 4).

**Mortality (deaths) rates analysis.** Breast cancer death rates were highest among the 70+ year age group, rising from 105.35 per 100,000 populations in 2000 to 135.60 in 2021. In contrast, for deaths due to cervical cancer, the highest age group changed from 65−69 years old in 2000–70+ years old in 2021, and the rates declined dramatically from 146.91 per 100,000 populations in 2000 to 84.51 in 2021. Death rates of ovarian cancer were also highest among the 70+ year age group, rising from 23.24 per 100,000 populations in 2000 to 31.24 in 2021. Finally, for uterine cancer, the highest death rate continued to be in the 70+ year age group, falling from 14.86 per 100,000 populations in 2000 to 12.84 in 2021 (Fig 5).

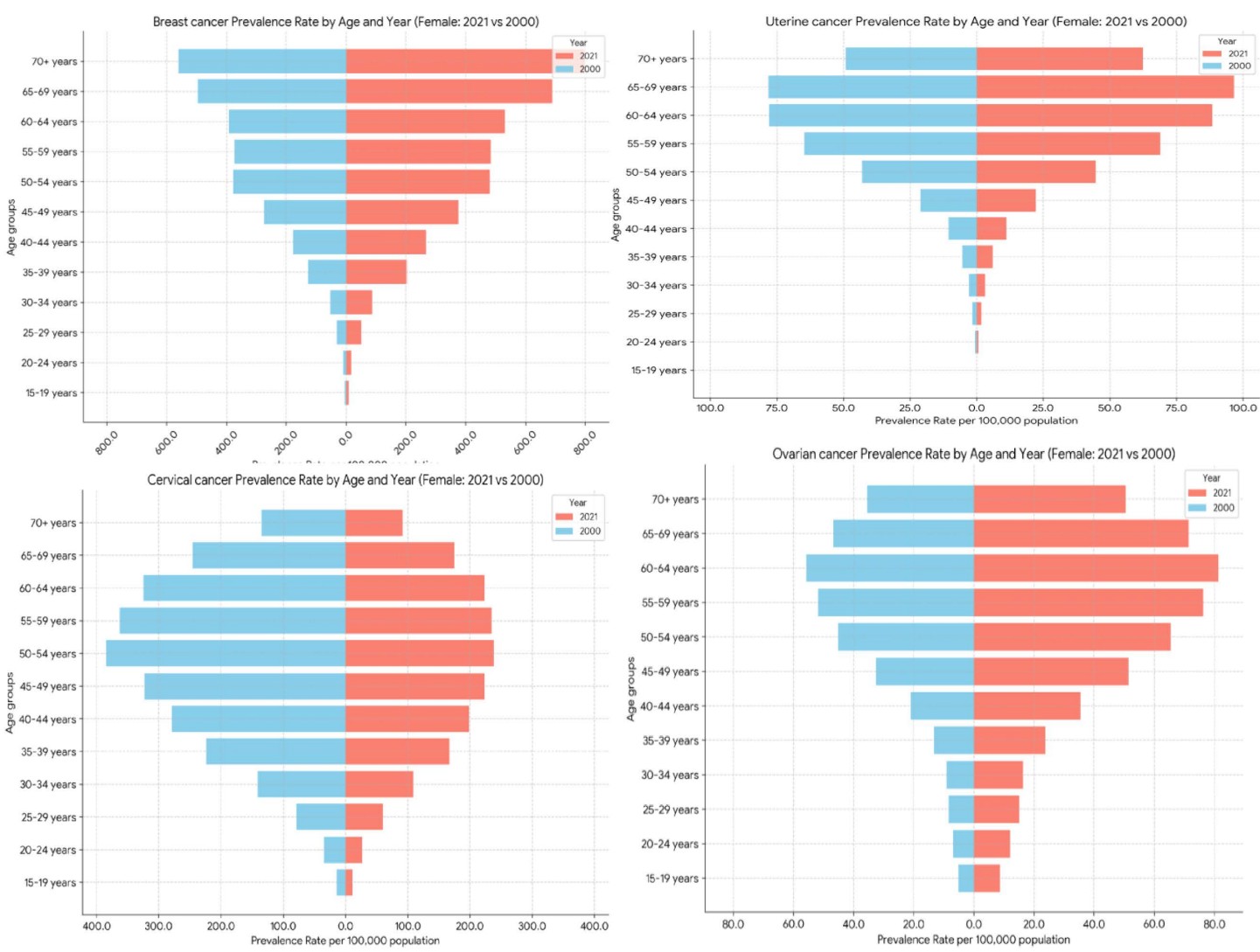

**Fig 3. Age-standardize prevalence of Breast, Cervical, Uterine and Ovarian cancer in Ethiopia.**

### Age-standardized burden of female breast, cervical, ovarian and uterine cancer in Ethiopia: A comparative analysis, 2000 and 2021

The breast cancer burden, quantified in DALYs, remarkably increased in the 75 + age group (29.97% rises) and modestly in the 15–49 age group (7.23% rise) between 2000 and 2021. There was a slight decrease in DALYs in the 50–74 age groups (−0.77%). There was a constant and notable increase in YLDs among all age groups, with the highest percentage changes in the 15–49 years (47.43%) age group and second highest in 75 + years (46.68%). YLLs also presented in the same trends as DALYs, with a notable increase in the 75 + year's age group (29.48%). This reflects an increasing burden of living with breast cancer among younger and older females.

Contrary to the sharp increase in breast cancer, the cervical cancer burden experienced a steep decrease in all age groups and in all three measures. DALYs due to cervical cancer registered steady decreases, with 50–74 age groups having the highest percentage decrease at −54.48%, followed by the 15–49 age groups at −53.90%. The same steep

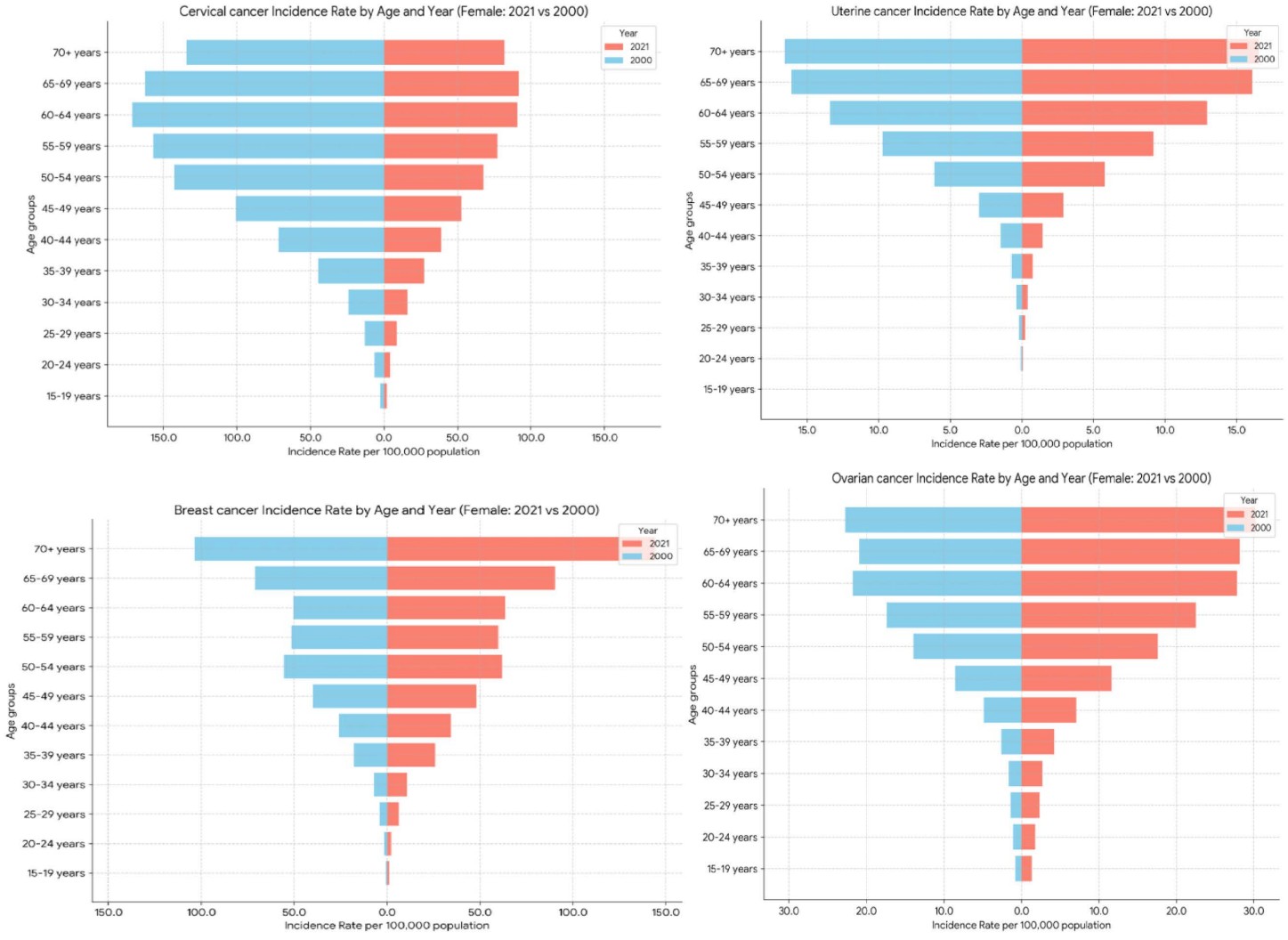

**Fig 4. Age-standardize Incidence of Breast, Cervical, Uterine and Ovarian cancer in Ethiopia.**

decreases were registered for YLDs (−43.54% in 50–74 years) and YLLs (e.g., −54.60% in 50–74 years). These results reflect a welcome change in the epidemiological trend of cervical cancer.

Uterus cancer also experienced an overall reduction in DALYs and YLLs across all age groups. The most significant reductions in DALYs and YLLs were in the 15–49 and 50–74 years age groups (approximately −27% to −28%). For YLDs, the changes were minimal, with a slight increase in the 15–49 and 75 + age groups and a slight decrease in the 50–74 age groups. The burden of ovarian cancer rose in every age group and for every measure (DALYs, YLDs, and YLLs). The 15–49 age groups had the largest percent rise in YLDs (58.25%) and DALYs (38.42%). The 75 + age group also experienced steep rises in DALYs (30.15%) and YLLs (30.05%) (Table 1). Trends indicate an increasing public health issue of ovarian cancer.

### The national burden prediction for female breast, cervical, uterine and ovarian cancer in Ethiopia to 2040

The ARIMA model projects a continued increasing trend in incidence rates for all analyzed female cancer types in Ethiopia from 2022 to 2040. The projected incidence rates (per 100,000 population) for each cancer type by 2040, along with their

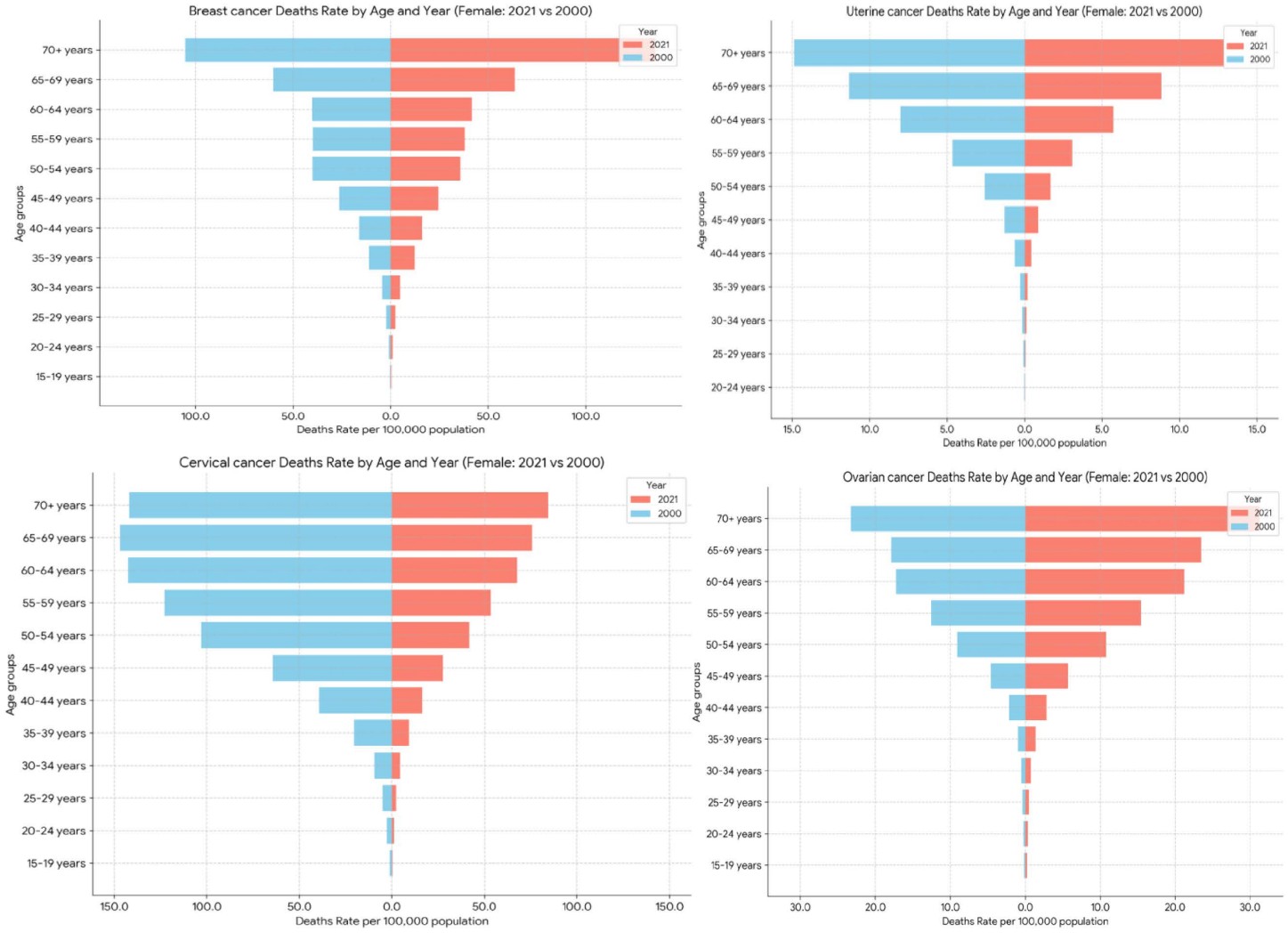

**Fig 5. Age-standardize mortality rate of Breast, Cervical, Uterine and Ovarian cancer in Ethiopia.**

95% uncertainty intervals (UI), which indicate the range within which the true future incidence is likely to fall and provide an important indication of the potential future burden of these diseases.

In 2021, the incidence of ovarian cancer was approximately 3.80 per 100,000. The incidence rate is projected to rise to approximately 5.80 (95% UI: 4.45–7.15) per 100,000 by 2040. This suggests a notable increase but also indicates a considerable range of possible outcomes. Although the 2021 incidence of cervical cancer was 14.59 per 100,000, its incidence is projected to increase significantly to approximately 21.10 (95% UI: 8.21–33.98) per 100,000. This wide interval highlights the higher uncertainty associated with the long-term forecast for this cancer type, despite the clear increasing trend. Breast cancer incidence is also projected to increase from 13.53 per 100,000 in 2021 to 20.66 (95% UI: 13.93–27.39) per 100,000 in 2040, indicating a substantial anticipated increase. For uterine cancer, the incidence in 2021 was approximately 1.24 per 100,000; however, in 2040 it is projected to rise to 1.84 (95% UI: 1.16–2.51) per 100,000, suggesting a more predictable, albeit increasing, trend compared to cervical cancer (Fig 6).

**Table 1. Age-standardized burden of female breast, cervical, ovarian and uterine Cancer in Ethiopia (2000 and 2021).**

| Cause | Measure | Age class | ASR (95% UI) in 2000 | ASR(95% UI) in 2021 | Absolute Change | Annual Percentage Change (%) |
|---|---|---|---|---|---|---|
| Breast cancer | DALYs | 15-49 years | 283.08 (189.73-411.95) | 303.56 (233.13-389.26) | 20.47 | 7.23 |
| | | 50-74 years | 1471.96 (1018.38-2106.47) | 1460.69 (1148.49-1854.18) | −11.27 | −0.77 |
| | | 75+years | 1602.75 (1173.46-2119.78) | 2083.16 (1697.83-2528.49) | 480.41 | 29.97 |
| | YLDs | 15-49 years | 5.50 (3.11-8.44) | 8.11 (5.19-11.82) | 2.61 | 47.43 |
| | | 50-74 years | 34.56 (21.62-52.62) | 43.19 (28.74-60.71) | 8.63 | 24.97 |
| | | 75+years | 46.19 (31.28-64.36) | 67.75 (47.39-88.88) | 21.56 | 46.68 |
| | YLLs | 15-49 years | 277.58 (186.12-404.44) | 295.45 (225.79-378.38) | 17.86 | 6.44 |
| | | 50-74 years | 1437.40 (996.37-2056.99) | 1417.51 (1115.02-1802.42) | −19.90 | −1.38 |
| | | 75+years | 1556.56 (1140.69-2056.55) | 2015.40 (1643.73-2449.76) | 458.84 | 29.48 |
| Cervical cancer | DALYs | 15-49 years | 645.59 (494.56-934.61) | 297.61 (208.99-459.31) | −347.98 | −53.90 |
| | | 50-74 years | 3935.62 (3015.53-5586.80) | 1791.67 (1318.57-2525.78) | −2143.95 | −54.48 |
| | | 75+years | 1816.55 (1292.62-2671.46) | 1134.13 (846.46-1528.98) | −682.42 | −37.57 |
| | YLDs | 15-49 years | 9.68 (6.31-14.36) | 6.58 (3.97-10.60) | −3.10 | −32.03 |
| | | 50-74 years | 44.81 (30.33-64.15) | 25.30 (16.40-39.77) | −19.51 | −43.54 |
| | | 75+years | 23.24 (14.28-34.59) | 15.76 (10.13-23.50) | −7.48 | −32.20 |
| | YLLs | 15-49 years | 635.91 (487.46-920.23) | 291.03 (204.86-449.56) | −344.88 | −54.23 |
| | | 50-74 years | 3890.81 (2979.37-5532.35) | 1766.38 (1300.26-2493.46) | −2124.43 | −54.60 |
| | | 75+years | 1793.31 (1277.20-2643.59) | 1118.38 (834.36-1505.45) | −674.93 | −37.64 |
| Uterine cancer | DALYs | 15-49 years | 10.79 (6.51-15.77) | 7.78 (4.68-14.66) | −3.01 | −27.86 |
| | | 50-74 years | 189.24 (120.53-270.51) | 136.96 (85.39-233.96) | −52.27 | −27.62 |
| | | 75+years | 226.02 (140.04-339.96) | 194.16 (121.99-300.22) | −31.86 | −14.10 |
| | YLDs | 15-49 years | 0.26 (0.14-0.42) | 0.27 (0.14-0.57) | 0.01 | 3.22 |
| | | 50-74 years | 5.41 (3.01-8.87) | 5.36 (3.05-10.25) | −0.06 | −1.04 |
| | | 75+years | 5.35 (2.92-8.91) | 5.79 (3.34-10.11) | 0.44 | 8.17 |
| | YLLs | 15-49 years | 10.53 (6.34-15.40) | 7.51 (4.51-14.15) | −3.01 | −28.64 |
| | | 50-74 years | 183.83 (117.02-261.32) | 131.61 (81.82-223.84) | −52.22 | −28.41 |
| | | 75+years | 220.66 (135.99-331.64) | 188.37 (117.92-291.14) | −32.29 | −14.64 |
| Ovarian cancer | DALYs | 15-49 years | 42.06 (22.90-74.52) | 58.23 (28.55-84.50) | 16.16 | 38.42 |
| | | 50-74 years | 424.40 (189.84-750.27) | 523.08 (258.83-752.73) | 98.68 | 23.25 |
| | | 75+years | 374.11 (131.76-623.87) | 486.92 (243.12-681.78) | 112.81 | 30.15 |
| | YLDs | 15-49 years | 1.11 (0.55-2.17) | 1.75 (0.90-2.77) | 0.64 | 58.25 |
| | | 50-74 years | 8.23 (3.55-15.79) | 10.78 (5.16-16.73) | 2.55 | 31.02 |
| | | 75+years | 7.41 (2.45-14.24) | 10.01 (4.76-16.29) | 2.60 | 35.07 |
| | YLLs | 15-49 years | 40.96 (22.36-72.40) | 56.48 (27.64-82.02) | 15.52 | 37.89 |
| | | 50-74 years | 416.17 (185.76-736.62) | 512.29 (254.12-736.01) | 96.13 | 23.10 |
| | | 75+years | 366.69 (128.83-610.22) | 476.90 (238.13-666.55) | 110.21 | 30.05 |

DALYs: Disability-Adjusted Life Years YLDs: Years Lived with Disability YLLs: Years of Life Lost.

## Percentage contribution of risk factors to all-age DALYs of breast, cervical, uterine, and ovarian cancer in Ethiopia in 2021

In 2021 a diet high in red meat was the leading risk factors for breast cancer account 11.34% (95% UI: 0.00–24.13) of cases followed by high alcohol use is associated with 2.65% (95% UI: 0.95–4.33) cases. Smoking contributes to 0.18% (95% UI: 0.12–0.25) of breast cancer cases and 1.41% (95% UI: 0.75–2.24) of cervical cancer cases. Second hand

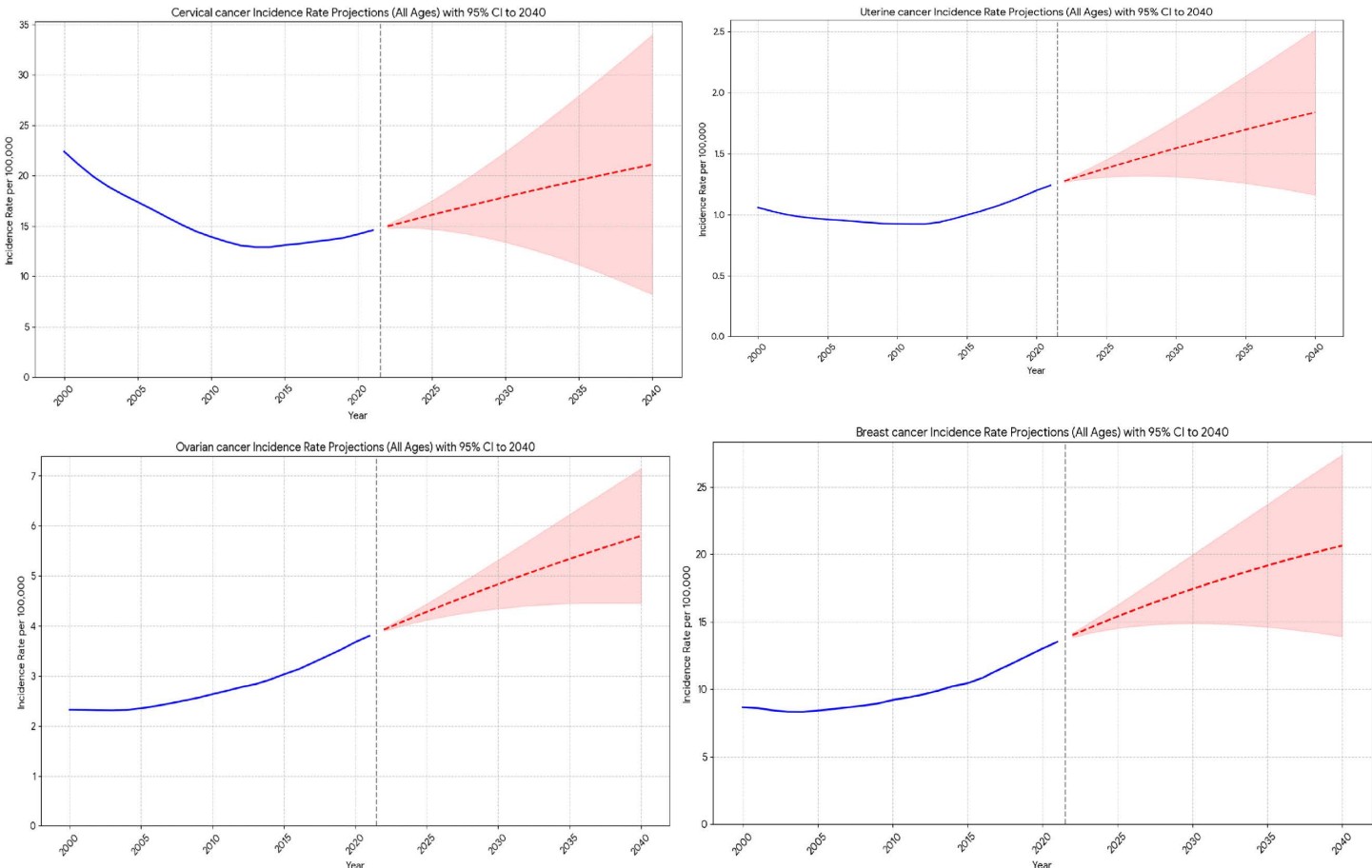

**Fig 6. Projections of the Incidence Rates for female specific cancer in Ethiopia from 2022 to 2040.**

smoke exposure is linked to 0.56% (95% UI: −0.13–1.31) of breast cancer cases. High fasting plasma glucose is associated with 1.66% (95% UI: −0.46–3.92) of breast cancer cases, while high body-mass index (BMI) is linked to 0.30% (95% UI: −0.54–1.18) of breast cancer cases, 2.01% (95% UI: 0.03–4.41) of ovarian cancer cases, and 12.53% (95% UI: 9.11–16.68) of uterine cancer cases. Low physical activity accounts for 1.24% (95% UI: 0.24–2.29) of breast cancer cases. Notably, unsafe sex is responsible for 100% (95% UI: 100–100) of cervical cancer cases, highlighting its dominant role in this cancer type.

## Discussion

This study critically examines the epidemiological landscape of common female breast, cervical, ovarian and uterine cancer in Ethiopia from 2000–2021, integrating insights from the national and sub-national (regional) trends in prevalence, incidence, mortality, DALYs, YLDs, and YLLs and associated risk factors. Additionally, it includes the projected incidence rate till 2040, providing comprehensive understanding of the evolving cancer burden and its implications for public health.

At the national level, the statistics reveal a clear trend: the prevalence of cancer among women is increasingly affecting the female population in Ethiopia. Most notable is the finding that breast cancer has seen a dramatic increase, marked by an increase in both the prevalence and the rate of new cases. This finding is consistent with an international trend, where more and more women in countries like Ethiopia are being diagnosed with breast cancer, perhaps due to changes

in lifestyle, diet, urbanization, and improvements in disease detection equipment [13–15]. The dramatic increase in YLDs related to breast cancer is particularly germane, as the result show that an increasing number of women are surviving with the condition; however, many struggle with the long-term effects of the disease as well [16]. This finding highlights an immediate need for continued, supportive care and services focused on helping women cope with their lives after diagnosis.

The trend of cervical cancer is slightly more complex. Although the incidence and prevalence of cervical cancer are increasing, the overall burden—measured in DALYs—is declining. This pattern aligns with findings from global studies [16,17]. The decline in DALYs is largely driven by a reduction in YLLs, which suggests that fewer women are dying prematurely from cervical cancer due to improvements in screening and treatment. However, the increase in YLDs highlights that many women continue to live with the long-term physical and psychological effects of cervical cancer. This underscores the importance of integrating palliative care and rehabilitation services to improve quality of life for these women and further strengthening the cervical cancer screening programs.

Amongst cancer affecting woman's health, ovarian cancer demonstrates truly alarming upward trends. We are witnessing dramatic increases in both incidence and prevalence alongside a stunning surge in the burden it places on women's health. Consistent finding were reported in global studies [18]. What is particularly striking at this stage is the rising trend in YLD, reflecting a rapidly escalating burden among affected populations. This sharp increase highlights the urgent need for improved early detection methods and more effective treatments, as ovarian cancer often goes undiagnosed until it advanced stages.

Alongside other cancer, the increase in YLDs with uterine cancer diagnosis is noticeably sharp. Similarly the global GBD result also showed the consistent increment of uterine cancer especially in aged greater than 55 years old [19,20]. This data showcases rising challenges that women face in managing a health condition—disability—after diagnosis. There is an apparent gap for enhanced recovery support and care to help them recover fully.

## Sub-national dynamics: understanding local realities

We found significant regional disparity in Ethiopian female cancer burden and its implications for the need to tailor cancer control to local conditions. The high-population Oromia and Amhara regions have a disproportionately high incidence and prevalence of female cancer, such as breast and cervical cancer. This is likely the consequence of a combination of high population density, limited access to early detection service, and potential environmental and behavioral risk factors that are present in these places. The stress on healthcare facilities in such high-density places could contribute to delayed diagnosis and treatment as well as to a rise in morbidity and mortality.

On the other hand, there is a distinct pattern of high breast and uterine cancer prevalence with significant rises in YLDs in Addis Ababa. Urban advantage in this setting—expressed as increased access to therapeutic and diagnostic services— would likely lead to earlier detection of cancer, which explains the augmented noticed prevalence as well as YLDs. It also indicates a hidden burden of chronic disability among survivors, touting the need for sustainable secondary care services rather than acute care. The reduction of cervical cancer DALYs in Addis Ababa highly likely reflects the efficacy of geographically targeted prevention, such as Human papilloma virus (HPV) vaccination initiatives and accessible screening programs and, signifying a model for urban health interventions.

Conversely, almost doubling of ovarian cancer burden in the SNNP region, in all areas, is a cause for concern. This dramatic increase may be attributed to area-specific determinants like limited access to health services, lower awareness, and potential differences in reproductive or lifestyle risk factors. The surge calls for immediate targeted epidemiologic research and resource allocation to address this growing public health threat.

These regional variations illuminate broader national challenges including uneven distribution of healthcare resources, disparities in health literacy, and socio-economic inequalities that impact cancer prevention and management. They highlight that a one-size-fits-all approach is inadequate for Ethiopia; instead, cancer control strategies must be regionally

adapted to address local epidemiology, healthcare capacity, and social determinants. Strengthening decentralized health systems with enhanced diagnostic, treatment, and supportive care tailored to regional needs is crucial to reduce disparities and improve outcomes for Ethiopian women across diverse settings.

### Age standardized burden: who is most affected?

Studying trend of cancer by age provides critical insights into which groups are most vulnerable. For breast cancer, the increasing DALYs and YLDs across various age groups, especially among younger women (15–49) and older women (75+), means that women of all ages are increasingly living with the disease and its impact. This calls for age-appropriate screening guidelines, compassionate support services, and awareness campaigns that recognize the unique life stages and challenges faced by younger women (like balancing cancer with family or career) and older women (who may have other health conditions). In a remarkable and promising shift, the impact of cervical cancer has lessened significantly among all age groups. The notable decline in DALYs and YLLs among women aged 15–49 and 50–74 indicates that public health initiatives, potentially HPV vaccination campaigns along with screening boosts, are truly saving lives and decreasing the burden of this disease [21]. This trend is inspiring as it highlights the benefits of targeted preventive measures coupled with early diagnostics.

Compared to this progress, the situation remains bleak for ovarian cancer, which is rising across all age groups. Younger working women (15–49 years) display the most pronounced increase in YLDs and DALYs associated with particular disability burdens. There is a striking need to focus on developing timely detection systems along with better treatment strategies for this aggressive form of cancer targeting younger populations [22]. While uterine cancer has seen overall reductions in its burden, a slight increase in YLDs in some age groups reminds us that comprehensive post-treatment care remains essential.

**Modifiable risk factors contributing to female cancer in Ethiopia in 2021.** According to the risk factor analysis, several modifiable factors contribute to the burden of female cancer in Ethiopia. A diet high in red meat and high alcohol use increase breast cancer risk mainly through hormonal and inflammatory pathways that promote tumor development [23,24]. Smoking and secondhand smoke exposure introduce carcinogens causing DNA damage, contributing to both breast and cervical cancer [23–25]. Unsafe sex is the predominant risk factor for cervical cancer, enabling persistent HPV infection, the primary cause of cervical neoplasia [25,26]. High fasting plasma glucose and elevated body mass index (BMI) are linked to breast, ovarian, and uterine cancer via mechanisms such as insulin resistance, chronic inflammation, and hormonal imbalances [23,24,27,28]. These factors show varying strength of association across different cancer but can be modified through targeted interventions including lifestyle and dietary counseling, smoking cessation programs, management of obesity and diabetes, promotion of physical activity, alcohol reduction strategies, as well as widespread HPV vaccination and safe sex education. Addressing these modifiable risks is critical for effective cancer prevention and reducing the future disease burden.

### Projection implications

The results of projection uniformly indicate a rising burden of female cancer in Ethiopia over the next two decades. Breast and ovarian cancer are expected to rise substantially. Uterine cancer, on the other hand, exhibits a more consistent but nonetheless raising its trend. Despite the encouraging past reductions in DALYs and YLLs, cervical cancer incidence is projected to increase significantly by 2040. The consistent upward trend across all four cancer types suggests that, if historical patterns persist, the demand for cancer diagnosis, treatment, and palliative care services will significantly increase.

This study, similar to other GBD analyses, has limitations stemming from data integrity and quality, particularly in low socioeconomic development regions including Ethiopia, which can lead to potential underestimations, misdiagnoses, and data heterogeneity. Reliance on simulation techniques in data-scarce areas compromises accuracy, and projections

based solely on historical trends may not fully capture dynamic disease changes or the evolving influence of unanalyzed or emerging risk factors. Furthermore, our analysis is based on population-level data, which precludes the exploration of individual-level risk factors and the differentiation of specific case types. We acknowledge that this limitation may influence the granularity of our findings; however, the GBD framework remains the most robust tool for providing a comprehensive, standardized, and longitudinal assessment of disease burden in data-scarce settings. lastly, the study's scope was limited to female breast, cervical, uterine, and ovarian cancer, excluding other female cancer types and specific tumor pathological features, and the risk factors considered were confined to those within the GBD database, potentially influencing the comprehensiveness of the findings and hindering precise prevention and control efforts.

Nonetheless, this study makes a significant contribution to Ethiopian and African cancer literature by providing the most comprehensive, nationally representative analysis of female cancer burden trends and projections through 2040. Using the standardized GBD framework, it offers consistent, comparable data over two decades that integrate mortality and disability measures (DALYs, YLLs, and YLDs). This robust, longitudinal design addresses key epidemiological gaps, delivering critical evidence to inform future research, policy, and targeted cancer control strategies in low-resource settings across the continent.

## Conclusion and recommendations

Ethiopia is faced with a dynamic and increasing burden of female cancer that is characterized by a substantial increase in breast and ovarian cancer incidence and disability in the past two decades and a projected increase in all female cancer types, including cervical cancer, by 2040. Although cervical cancer mortality has declined, the overall burden of women living with cancer-related disabilities highlights significant unmet needs in supportive care. Addressing this multifaceted challenge requires a comprehensive national cancer control strategy that is contextualized regionally. We recommend a heightened emphasis on prevention and early detection, with specific, actionable steps such as introducing HPV self-sampling programs to increase cervical cancer screening uptake and implementing national ovarian, breast and uterine cancer screening pilots, particularly in regions with the highest burden. This strategy must also include expanded capacity for diagnosis and treatment, comprehensive palliative care, and ongoing investment in policy and research to protect women's health in Ethiopia. These efforts directly contribute to achieving Sustainable Development Goal 3.4, which aims to reduce premature mortality from non-communicable diseases by one-third by 2030, underscoring the critical role of cancer control in meeting global health targets.

## Supporting information

**S1 File.   Supplemental fig 1.** Sub-national prevalence, incidence and mortality rate of breast, cervical, ovarian and uterine cancer in Ethiopia (2000–2021). **Supplemental fig 2.** the Sub-national DALYs, YLD, and YLLs trend of Female specific cancer in Ethiopia (2000–2021).
(DOCX)

## Author contributions

**Conceptualization:** Molalign Aligaz Adisu, Tesfaye Engdaw Habtie, Tegene Atamenta Kitaw.

**Data curation:** Molalign Aligaz Adisu, Bogale Molla Woreta, Alemu Birara Zemariam.

**Formal analysis:** Molalign Aligaz Adisu, Tesfaye Engdaw Habtie, Tegene Atamenta Kitaw, Abraham Dessie Gessesse, Bogale Molla Woreta, Yabibal Asfaw Derso, Alemu Birara Zemariam.

**Funding acquisition:** Molalign Aligaz Adisu.

**Investigation:** Molalign Aligaz Adisu.

**Methodology:** Molalign Aligaz Adisu, Tesfaye Engdaw Habtie, Tegene Atamenta Kitaw, Abraham Dessie Gessesse, Bogale Molla Woreta, Yabibal Asfaw Derso, Alemu Birara Zemariam.

**Project administration:** Molalign Aligaz Adisu.

**Resources:** Molalign Aligaz Adisu.

**Software:** Molalign Aligaz Adisu, Tesfaye Engdaw Habtie.

**Supervision:** Molalign Aligaz Adisu.

**Validation:** Molalign Aligaz Adisu.

**Visualization:** Molalign Aligaz Adisu.

**Writing – original draft:** Molalign Aligaz Adisu, Tesfaye Engdaw Habtie, Tegene Atamenta Kitaw, Abraham Dessie Gessesse, Bogale Molla Woreta, Yabibal Asfaw Derso, Alemu Birara Zemariam.

**Writing – review & editing:** Molalign Aligaz Adisu, Tesfaye Engdaw Habtie, Tegene Atamenta Kitaw, Abraham Dessie Gessesse, Bogale Molla Woreta, Yabibal Asfaw Derso, Alemu Birara Zemariam.

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
