## [Decision Letter · Decision Letter 0]

22 Aug 2025

PONE-D-25-35361The Rising Burden of Female Cancers in Ethiopia: A 21-years National and Sub-national Analysis of Breast, Cervical, Ovarian and Uterine Cancers (2000–2021) with Projections to 2040 – Findings from the Global Burden of Disease StudyPLOS ONE

Dear Dr. Aligaz,

Thank you for submitting your manuscript to PLOS ONE. After careful consideration, we feel that it has merit but does not fully meet PLOS ONE’s publication criteria as it currently stands. Therefore, we invite you to submit a revised version of the manuscript that addresses the points raised during the review process.

**ACADEMIC EDITOR: The topic is of interest. This manuscript might contribute significantly to the current literature of Ethiopia. Please address all comments from reviewers. **

We look forward to receiving your revised manuscript.

Kind regards,

Thien Tan Tri Tai Truyen, M.D.

Academic Editor

PLOS ONE

Journal Requirements:

Reviewers' comments:

Reviewer's Responses to Questions

**Comments to the Author**

1. Is the manuscript technically sound, and do the data support the conclusions?

Reviewer #1: Yes

Reviewer #2: Partly

Reviewer #3: Yes

Reviewer #4: Yes

2. Has the statistical analysis been performed appropriately and rigorously? 

Reviewer #1: Yes

Reviewer #2: Yes

Reviewer #3: Yes

Reviewer #4: Yes

3. Have the authors made all data underlying the findings in their manuscript fully available?

Reviewer #1: Yes

Reviewer #2: Yes

Reviewer #3: Yes

Reviewer #4: Yes

4. Is the manuscript presented in an intelligible fashion and written in standard English?

Reviewer #1: Yes

Reviewer #2: No

Reviewer #3: Yes

Reviewer #4: Yes

5. Review Comments to the Author

Reviewer #1: The manuscript was presented and written in standard English with no major errors which caused difficulties to understand, however I recognized some minor grammar mistakes that should be reviewed and corrected. All the grammar errors were marked and recommendations were noted in the reviewed version attached. Moreover, the manuscript mostly mentioned and discussed about screening along with early detection of the disorders and lacked discussion regarding primary prevention such as intervention to prevent risk factors of the diseases. Primary prevention should be mentioned more in the manuscript.

Please refer to this article for reference:

https://doi.org/10.1016/j.xagr.2025.100526

Reviewer #2: This study analyzed data from the Global Burden of Disease (GBD) 2021 to highlight the 21-year trends and future projections (up to 2040) of four primary female cancers—breast, cervical, ovarian, and uterine—in Ethiopia. It presents comprehensive insights into national and sub-national trends across incidence, prevalence, mortality, Years Lived with Disability (YLDs), Years of Life Lost (YLLs), and Disability-Adjusted Life Years (DALYs). Key findings indicate significant increases in breast and ovarian cancer burdens, while cervical cancer, despite reduced mortality, shows a rise in associated disability.

This study demonstrates notable strengths. The study provides a comprehensive national and sub-national analysis using large-scale GBD 2021 data, supported by long-term trends and ARIMA modeling for reliable projections. The consistent rise in all four cancer types indicates a likely increase in demand for diagnosis, treatment, and palliative care if current patterns continue. However, This study shares common GBD limitations, including heterogeneity data quality in low socioeconomic development regions and reliance on simulations, which may affect accuracy. Moreover, the analysis is based on population-level data, lacking individual-level information and case-type differentiation.

1. Overall

1.1. The manuscript should be thoroughly proofread to ensure correct grammar, spelling, academic language, and adherence to formatting in accordance with the author's guidelines.

1.2. Abbreviations should be defined in full upon their first appearance in the manuscript and should not be redefined in subsequent sections.

2. Abstract

2.1. The abstract should not exceed 300 words.

2.2. The Methods section of the abstract should be carefully reviewed to ensure clarity, semantic completeness, and proper capitalization of all proper nouns.

2.3. The Results and Discussion sections should be reviewed and organized in a clear and consistent manner to enhance readability, with recommendations presented according to each disease group.

3. Introduction

3.1. Typos should be carefully checked, and sentences should be written clearly, comprehensively, and coherently to enhance clarity and readability.

3.2. The Introduction should provide a description of each sub-national region analyzed, along with a clear rationale for their selection—indicating whether they are representative of the broader region or the national population. If applicable, the characteristics and their influence on disease burden should be clearly outlined.

4. Methods

4.1. The Methods section should be supplemented with appropriate references and citations for the methodologies applied in the study.

4.2. The final paragraph beginning with “This method aims to lay the foundation...” functions as a conclusion and should be moved to the Discussion section rather than remaining in the Methods.

5. Results

5.1. This section should be revised using varied and clear descriptions to enhance readability and logical flow. Abbreviations previously defined may be used where appropriate. Furthermore, this part should focus solely on presenting the results without interpretation. The authors' comments or explanations at the end of each subsection should be moved to the Discussion section.

5.2. In the subsection “ The national prevalence, incidence and death Trends of Female breast, cervical, ovarian and uterine cancer in Ethiopia (2000-2021)”, the mortality is not addressed and should be included. While cervical cancer shows an overall increase in prevalence and incidence, one segment demonstrates a decline—this exception should be clearly identified and discussed.

5.3. In the subsection “ The national prevalence, incidence and death Trends of Female breast, cervical, ovarian and uterine cancer in Ethiopia (2000-2021)”, the sentence “ Lastly, uterine cancer incidence nearly double,…” is unclear or inappropriate and should be reviewed and revised for accuracy and clarity.

5.4. The subsection “ Sub-National prevalence, Incidence and mortality trends of beast, cervical, ovarian and uterine cancer in Ethiopia (2000-2021)” should place greater emphasis on describing mortality patterns, rather than focusing primarily on prevalence and incidence.

5.5. The subsection “ Sub-National prevalence, Incidence and mortality trends of beast, cervical, ovarian and uterine cancer in Ethiopia (2000-2021)”, part “ Cervical cancer” : Regional variations in trends should be further elaborated. Although cervical cancer incidence and prevalence increased and mortality declined overall, some regions deviated from this pattern. These discrepancies should be clearly described and analyzed.

5.6. The subsection “ Sub-National prevalence, Incidence and mortality trends of beast, cervical, ovarian and uterine cancer in Ethiopia (2000-2021)”, part “ Ovarian cancer” : Even though incidence, prevalence, and mortality increased across regions, the extent of these increases varied notably. The results should emphasize these regional differences to better understand the disparities.

5.7. In the subsection “Deconstructing the Total Health Loss: Disability-Adjusted Life Years, Years Lived with Disability, and Years of Life Lost for Female Breast, Cervical, Ovarian, and Uterine Cancers in Ethiopia (2000–2021)”, the title “Cervical cancer: a complex picture of decreasing mortality but rising disability” is not yet supported by a clear and accurate description of the corresponding data trends. While the title suggests rising disability, the opening sentence of the paragraph appears inconsistent with the reported data. Specifically, from approximately 2000 to 2021, DALYs decreased sharply, YLDs increased slightly, and YLLs declined marginally.

5.8. The subsection titled “Age-standardized prevalence, incidence, and mortality rate of breast, cervical, uterine, and ovarian cancer in Ethiopia” should be placed before the subsection “Age-standardized burden of female breast, cervical, ovarian, and uterine cancers in Ethiopia: A Comparative Analysis, 2000 and 2021” to ensure a more logical and coherent flow of information.

6. Discussion

6.1. The Discussion section should be reviewed and rewritten in an academic style, ensuring that interpretive or clarifying comments—currently placed in the Results section—are appropriately relocated. Revisions should also reflect any updates to the Results section, with careful attention to spelling and grammar accuracy. Furthermore, the Discussion section should include more in-depth explanations and supporting arguments to better interpret and contextualize the results.

6.2 The third paragraph beginning with “The trend of cervical cancer is slightly more complex…” ; the sentences: “This result was consistent with the global studies specifically in breast cancer (14, 15). This is primarily because YLLs have decreased, indicating that those women who were dying from the disease are not losing lives to it because screening and treatment options are improving.” This statement is confusing and should be carefully reviewed and revised for clarity, accuracy, and logical consistency.

6.3. The subsection titled “Sub-National Dynamics: Understanding Local Realities” should emphasize the key disease patterns in each region, providing detailed explanations for these trends. It should also discuss the underlying regional characteristics and how they reflect broader national challenges and health system issues.

6.4 The final paragraph beginning with “According to the risk factor analysis result…” in the subsection “Age-Standardized Burden: Who is Most Affected?” could be improved by briefly outlining the underlying mechanisms through which the identified risk factors influence disease burden. Additionally, it should clarify how these factors may be modified through targeted interventions.

Reviewer #3: The manuscript explores increasing burden of female cancers in Euthopia. The manuscript is technically sound and well writtten. In My opininon authors should make following ammendement in their manuscript.

Authors should make a comparative study about increasing burden of female cancers around the Globe initially, in the introduction section and then proceed towards Euthopia.

Reviewer #4: The manuscript titled “The Rising Burden of Female Cancers in Ethiopia: A 21-years National and Sub national Analysis of Breast, Cervical, Ovarian and Uterine Cancers (2000–2021) with Projections to 2040 – Findings from the Global Burden of Disease Study” addresses a major public health issue in Ethiopia using nationally representative data from the Global Burden of Disease (GBD) 2021, which aligns with PLOS ONE’s focus on broad, impactful research. However, the manuscript requires major revisions before it can be considered for publication in peer-reviewed journal.

6. PLOS authors have the option to publish the peer review history of their article (what does this mean? ). If published, this will include your full peer review and any attached files.

**Do you want your identity to be public for this peer review?** For information about this choice, including consent withdrawal, please see our Privacy Policy .

Reviewer #1: **Yes: ** Thao-Ngan Nguyen Pham

Reviewer #2: **Yes: ** Bao Huy Le

Reviewer #3: No

Reviewer #4: No

---

## [Author Response · Author response to Decision Letter 1]

27 Aug 2025

Point-by-point response

Subject: submission of revised manuscript

Manuscript ID: PONE-D-25-35361

Title: The Rising Burden of Female Cancer in Ethiopia (2000–2021) and Projections to 2040: Insights from the Global Burden of Disease Study

To: PLOS ONE

Dear editor/reviewers.

Dear, all

We sincerely thank you for your detailed, insightful feedback and constructive comments on our manuscript. Your suggestions have been invaluable in enhancing the substance and content of our work.

We have carefully considered each comment and question rose by the each reviewer and editor has provided point-by-point responses, which are detailed on the following pages. Additionally, we have incorporated these changes into the revised manuscript. The manuscript has been reviewed by language professionals, and we have ensured compliance with the journal's guidelines.

Reviewer #1

The manuscript was presented and written in Standard English with no major errors which caused difficulties to understand, however I recognized some minor grammar mistakes that should be reviewed and corrected. All the grammar errors were marked and recommendations were noted in the reviewed version attached. Moreover, the manuscript mostly mentioned and discussed about screening along with early detection of the disorders and lacked discussion regarding primary prevention such as intervention to prevent risk factors of the diseases. Primary prevention should be mentioned more in the manuscript.

Please refer to this article for reference: https://doi.org/10.1016/j.xagr.2025.100526

Response:

We thank the reviewer #1 for the careful reading and insightful feedback on our manuscript. We appreciate the positive note regarding the overall clarity and use of Standard English. We have addressed the minor grammar errors as marked in the reviewed version to ensure improved accuracy and readability throughout the manuscript.

Regarding the comment on primary prevention, we agree that emphasizing interventions targeting modifiable risk factors is essential. Accordingly, we have expanded the discussion section to incorporate more comprehensive coverage of primary prevention strategies, including public health interventions aimed at reducing exposure to key risk factors associated with female cancers in Ethiopia such as smoking cessation, reduction of high alcohol consumption, promotion of healthy diet and physical activity, and awareness campaigns addressing unsafe sexual practices linked to cervical cancer.

This addition complements our existing discussion on screening and early detection, offering a more holistic view of cancer control approaches necessary for effective reduction in female cancer burden in Ethiopia. We believe these revisions significantly strengthen the manuscript’s relevance and utility for policymakers and public health practitioners.

Thank you again for your valuable comments.

Reviewer #2

This study analyzed data from the Global Burden of Disease (GBD) 2021 to highlight the 21-year trends and future projections (up to 2040) of four primary female cancers—breast, cervical, ovarian, and uterine—in Ethiopia. It presents comprehensive insights into national and sub-national trends across incidence, prevalence, mortality, Years Lived with Disability (YLDs), Years of Life Lost (YLLs), and Disability-Adjusted Life Years (DALYs). Key findings indicate significant increases in breast and ovarian cancer burdens, while cervical cancer, despite reduced mortality, shows a rise in associated disability.

This study demonstrates notable strengths. The study provides a comprehensive national and sub-national analysis using large-scale GBD 2021 data, supported by long-term trends and ARIMA modeling for reliable projections. The consistent rise in all four cancer types indicates a likely increase in demand for diagnosis, treatment, and palliative care if current patterns continue. However, this study shares common GBD limitations, including heterogeneity data quality in low socioeconomic development regions and reliance on simulations, which may affect accuracy. Moreover, the analysis is based on population-level data, lacking individual-level information and case-type differentiation.

Response to Reviewer

Dear Reviewer #2, Thank you for your valuable feedback. We agree with your assessment regarding the use of population-level data. As you correctly noted, our analysis is based on GBD data, which, by its nature, is at the population level. This limitation, which we have now explicitly stated in our revised manuscript, means we were unable to analyze individual-level information and differentiate specific case types.

While this does constrain the scope of our analysis to broad trends rather than individual risk factors, the GBD methodology was the most appropriate approach for our study's objective. It allowed us to provide a standardized, comprehensive, and longitudinal overview of the rising burden of female cancers in Ethiopia, which is a crucial first step in a region with very limited cancer data. Your point has been well-taken, and we believe the clarification in the revised manuscript strengthens the transparency of our study's methodology and limitations.

Comments

1. Overall

1.1. The manuscript should be thoroughly proofread to ensure correct grammar, spelling, academic language, and adherence to formatting in accordance with the author's guidelines.

1.2. Abbreviations should be defined in full upon their first appearance in the manuscript and should not be redefined in subsequent sections.

Response

Dear Reviewer, Thank you for your valuable feedback. We have addressed both of your comments. First, we conduct a thorough proofreading of the manuscript to ensure correct grammar, spelling, academic language, and adherence to the specified author guidelines for formatting. Second, we will ensure that all abbreviations are defined in full upon their first appearance in the manuscript and are not redefined in subsequent sections. We appreciate you bringing these points to our attention, as they will significantly improve the quality and clarity of our manuscript.

Comments:

2. Abstract

2.1. The abstract should not exceed 300 words.

2.2. The Methods section of the abstract should be carefully reviewed to ensure clarity, semantic completeness, and proper capitalization of all proper nouns.

Response:

Dear Reviewer, Thank you for your feedback regarding the Methods section of our abstract. We have carefully reviewed and revised the section to address your comments.

Comment

2.3. The Results and Discussion sections should be reviewed and organized in a clear and consistent manner to enhance readability, with recommendations presented according to each disease group.

Response

Thank you again for your valuable feedback. The results and discussion sections have been reorganized and now present the findings and recommendations by each disease group to improve clarity and readability.

Comments

3. Introduction

3.1. Typos should be carefully checked, and sentences should be written clearly, comprehensively, and coherently to enhance clarity and readability.

3.2. The Introduction should provide a description of each sub-national region analyzed, along with a clear rationale for their selection—indicating whether they are representative of the broader region or the national population. If applicable, the characteristics and their influence on disease burden should be clearly outlined.

Response:

We appreciate the reviewer’s advice regarding language clarity and typos. We have carefully proofread the entire manuscript and revised sentences to ensure they are clear, comprehensive, and coherent, significantly improving readability throughout the text.

Regarding the Introduction, we have included all sub-national regions and city administrations of Ethiopia during the period from 2000 to 2021 in our analysis. We explicitly state that all these regions were included to provide a comprehensive national picture rather than a selected sample. Additionally, we briefly outline important regional characteristics (e.g., population density, urbanization, healthcare access) that may influence the cancer burden, providing clear rationale for their inclusion. This contextual information enhances understanding of regional disparities in female cancer burden within Ethiopia.

Comment

4. Methods

4.1. The Methods section should be supplemented with appropriate references and citations for the methodologies applied in the study.

Response

Thank you for this feedback. We have reviewed the Methods section and have now added appropriate references and citations for all methodologies used in the study. The revised manuscript reflects these changes and is ready for your review.

Comment

4.2. The final paragraph beginning with “This method aims to lay the foundation...” functions as a conclusion and should be moved to the Discussion section rather than remaining in the Methods.

Response

We have moved the paragraph beginning with "This method aims to lay the foundation..." from the Methods section to the Discussion section as suggested. The revised manuscript is ready for your review.

Comment

5. Results

5.1. This section should be revised using varied and clear descriptions to enhance readability and logical flow. Abbreviations previously defined may be used where appropriate. Furthermore, this part should focus solely on presenting the results without interpretation. The authors' comments or explanations at the end of each subsection should be moved to the Discussion section.

Response

Thanks for the feedback. We have revised the Results section to improve readability and logical flow using varied and clear descriptions. We have also moved all interpretations and explanations to the Discussion section, ensuring the Results section focuses solely on presenting the data. The revised manuscript is ready for your review.

Comment

5.2. In the subsection “The national prevalence, incidence and death Trends of Female breast, cervical, ovarian and uterine cancer in Ethiopia (2000-2021)”, the mortality is not addressed and should be included. While cervical cancer shows an overall increase in prevalence and incidence, one segment demonstrates a decline—this exception should be clearly identified and discussed.

Response

Thank you for your feedback. We have revised the subsection to include the mortality trends. We have also explicitly identified and discussed the specific segment of cervical cancer that shows a decline in prevalence and incidence, as you suggested. The updated manuscript is ready for your review.

5.3. In the subsection “The national prevalence, incidence and death Trends of Female breast, cervical, ovarian and uterine cancer in Ethiopia (2000-2021)”, the sentence “Lastly, uterine cancer incidence nearly double…” is unclear or inappropriate and should be reviewed and revised for accuracy and clarity.

Response:

Thank you for your feedback. We have reviewed the sentence.

5.4. The subsection “Sub-National prevalence, Incidence and mortality trends of beast, cervical, ovarian and uterine cancer in Ethiopia (2000-2021)” should place greater emphasis on describing mortality patterns, rather than focusing primarily on prevalence and incidence.

Response:

Thank you for the feedback. We have revised the subsection to include and focus on mortality patterns. The revised manuscript is ready for your review.

Comment

5.5. The subsection “Sub-National prevalence, Incidence and mortality trends of beast, cervical, ovarian and uterine cancer in Ethiopia (2000-2021)”, part “Cervical cancer”: Regional variations in trends should be further elaborated. Although cervical cancer incidence and prevalence increased and mortality declined overall, some regions deviated from this pattern. These discrepancies should be clearly described and analyzed.

Response:

Thank you for your feedback. We have revised the subsection to further elaborate on regional variations. As you suggested, we have clearly described the specific regions that deviated from the overall national trend of increasing incidence and prevalence and declining mortality in discussion part. The revised manuscript is ready for your review.

Comments

5.6. The subsection “Sub-National prevalence, Incidence and mortality trends of beast, cervical, ovarian and uterine cancer in Ethiopia (2000-2021)”, part “Ovarian cancer”: Even though incidence, prevalence, and mortality increased across regions, the extent of these increases varied notably. The results should emphasize these regional differences to better understand the disparities.

Response:

Thank you for your valuable feedback. We have revised the manuscript to incorporate a more detailed analysis of the sub-national trends for ovarian cancer, emphasizing the regional disparities as you suggested. We believe these revisions significantly strengthen the manuscript.

Comment

5.7. In the subsection “Deconstructing the Total Health Loss: Disability-Adjusted Life Years, Years Lived with Disability, and Years of Life Lost for Female Breast, Cervical, Ovarian, and Uterine Cancers in Ethiopia (2000–2021)”, the title “Cervical cancer: a complex picture of decreasing mortality but rising disability” is not yet supported by a clear and accurate description of the corresponding data trends. While the title suggests rising disability, the opening sentence of the paragraph appears inconsistent with the reported data. Specifically, from approximately 2000 to 2021, DALYs decreased sharply, YLDs increased slightly, and YLLs declined marginally.

Response:

Thank you for your valuable feedback. We have carefully reviewed your comment and revised the manuscript accordingly. The new title for the subsection is "Cervical cancer: decreasing mortality alongside modest rise in disability," which we believe more accurately reflects the data trends. The revised paragraph now provides a clear and consistent description of the data, highlighting the decrease in total DALYs and YLLs while acknowledging the modest rise in YLDs. We are confident that these changes address your concerns and provide a more precise representation of our findings.

Comment

5.8. The subsection titled “Age-standardized prevalence, incidence, and mortality rate of breast, cervical, uterine, and ovarian cancer in Ethiopia” should be placed before the subsection “Age-standardized burden of female breast, cervical, ovarian, and uterine cancers in Ethiopia: A Comparative Analysis, 2000 and 2021” to ensure a more logical and coherent flow of information.

Response:

Thank you for the feedback. We have reordered the manuscript sections as you suggested improving the logical flow.

Comment

6. Discussion

6.1. The Discussion section should be reviewed and rewritten in an academic style, ensuring that interpretive or clarifying comments—currently placed in the Results section—are appropriately relocated. Revisions should also reflect any updates to the Results section, with careful attention to spelling and grammar accuracy. Furthermore, the Discussion section should include more in-depth explanations and supporting arguments to better interpret and contextualize the results.

6.2 The third paragraph beginning with “The trend of cervical cancer is slightly more complex…” the sentences: “This result was consistent with the global studies specifically in breast cancer (14, 15). This is primarily because YLLs have decreased, indicating that those women who were dying from the disease are not losing lives to it because screening and treatment options are improving.” This statement is confusing and should be carefully reviewed and revised for clarity, accuracy, and logical consistency.

Response:

Thank you for your comment. We have revised the paragraph to clarify that the decline in DALYs is due to reduced YLLs from better screening and treatment, while the increase in YLDs reflects ongoing disability, highlighting the need for supportive care.

Comment:

6.3. The subsection titled “Sub-National Dynamics: Understanding Local Realities” should emphasize the key disease patterns in each region, providing detailed explanations for these trends. It should also discuss the underlying regional characteristics and how they ref

---

## [Decision Letter · Decision Letter 1]

17 Sep 2025

PONE-D-25-35361R1The Rising Burden of Female Cancer in Ethiopia (2000–2021) and Projections to 2040: Insights from the Global Burden of Disease StudyPLOS ONE

Dear Dr. Aligaz,

Thank you for submitting your manuscript to PLOS ONE. After careful consideration, we feel that it has merit but does not fully meet PLOS ONE’s publication criteria as it currently stands. Therefore, we invite you to submit a revised version of the manuscript that addresses the points raised during the review process.

**ACADEMIC EDITOR: Thank you for addressing all comments. A minor revision to address the final comment is needed. **==============================

We look forward to receiving your revised manuscript.

Kind regards,

Thien Tan Tri Tai Truyen, M.D.

Academic Editor

PLOS ONE

Journal Requirements:

Reviewers' comments:

Reviewer's Responses to Questions

**Comments to the Author**

1. If the authors have adequately addressed your comments raised in a previous round of review and you feel that this manuscript is now acceptable for publication, you may indicate that here to bypass the “Comments to the Author” section, enter your conflict of interest statement in the “Confidential to Editor” section, and submit your "Accept" recommendation.

Reviewer #1: All comments have been addressed

Reviewer #2: (No Response)

Reviewer #3: All comments have been addressed

2. Is the manuscript technically sound, and do the data support the conclusions?

Reviewer #1: Yes

Reviewer #2: Yes

Reviewer #3: Yes

3. Has the statistical analysis been performed appropriately and rigorously? 

Reviewer #1: Yes

Reviewer #2: Yes

Reviewer #3: Yes

4. Have the authors made all data underlying the findings in their manuscript fully available?

Reviewer #1: Yes

Reviewer #2: Yes

Reviewer #3: Yes

5. Is the manuscript presented in an intelligible fashion and written in standard English?

Reviewer #1: Yes

Reviewer #2: Yes

Reviewer #3: Yes

6. Review Comments to the Author

Reviewer #1: (No Response)

Reviewer #2: The abstract still exceeds the 300-word limit and must be shortened to comply with PLOS ONE's guidelines.

The revisions to another section have been completed to satisfy the comments previously provided.

Reviewer #3: Accept, all comments have been adressed, can move towards publication................................

7. PLOS authors have the option to publish the peer review history of their article (what does this mean? ). If published, this will include your full peer review and any attached files.

**Do you want your identity to be public for this peer review?** For information about this choice, including consent withdrawal, please see our Privacy Policy .

Reviewer #1: **Yes: ** Thao-Ngan Nguyen Pham

Reviewer #2: **Yes: ** Bao Huy Le

Reviewer #3: No

---

## [Author Response · Author response to Decision Letter 2]

17 Sep 2025

Point by point response

Manuscript ID: PONE-D-25-35361R1

Title: The Rising Burden of Female Cancer in Ethiopia (2000–2021) and Projections to 2040: Insights from the Global Burden of Disease Study

Dear Dr. Thien Tan Tri Tai Truyen,

Thank you for the opportunity to revise our manuscript. We are grateful to you and the reviewers for the constructive feedback. We have carefully addressed all the comments raised during the review process. We have also included a marked-up copy of the manuscript showing the changes.

Please find our point-by-point response to the reviewer comments below.

Reviewer #1, Reviewer #3, and Academic Editor Comments

Comment: The manuscript has been reviewed and found to have adequately addressed all comments. It is now acceptable for publication with a minor revision.

Response: We sincerely thank the reviewers and the editor for their positive feedback. We are pleased that the previous revisions have successfully addressed their concerns and that the manuscript is now in a state of readiness for publication, pending one final change.

Reviewer #2 Comments

Comment: The abstract still exceeds the 300-word limit and must be shortened to comply with PLOS ONE's guidelines.

Response: We appreciate the reviewer for highlighting this critical point. We apologize for the oversight in our previous revision. We have now meticulously revised the abstract to ensure it adheres to the 300-word limit as required by the journal's guidelines. The revised abstract concisely presents the study's key findings while maintaining all essential information.

We believe these revisions have significantly improved the manuscript and that it now fully meets the publication criteria of PLOS ONE. We look forward to your positive response.

Sincerely,

Molalign Aligaz Adisu and Co-authors

---

## [Editor Report · Decision Letter 2]

18 Sep 2025

The Rising Burden of Female Cancer in Ethiopia (2000–2021) and Projections to 2040: Insights from the Global Burden of Disease Study

PONE-D-25-35361R2

Dear Dr. Aligaz,

We’re pleased to inform you that your manuscript has been judged scientifically suitable for publication and will be formally accepted for publication once it meets all outstanding technical requirements.

Kind regards,

Thien Tan Tri Tai Truyen, M.D.

Academic Editor

PLOS ONE
---

## [Editor Report · Acceptance letter]

PONE-D-25-35361R2

PLOS ONE

Dear Dr. Adisu,

I'm pleased to inform you that your manuscript has been deemed suitable for publication in PLOS ONE. Congratulations! Your manuscript is now being handed over to our production team.

Kind regards,

on behalf of

Dr. Thien Tan Tri Tai Truyen

Academic Editor

PLOS ONE